# Left-Handed Metamaterial-Inspired Unit Cell for S-Band Glucose Sensing Application

**DOI:** 10.3390/s19010169

**Published:** 2019-01-05

**Authors:** Mohammad Tariqul Islam, Ahasanul Hoque, Ali F. Almutairi, Nowshad Amin

**Affiliations:** 1Centre of Advanced Electronic and Communication Engineering, Faculty of Engineering and Built Environment, Universiti Kebangsaan Malaysia, 43600 Bangi, Selangor, Malaysia; 2Electrical Engineering Department, Kuwait University, Kuwait City 13060, Kuwait; 3Institute of Sustainable Energy (ISE), Universiti Tenaga Nasional, Jalan IKRAM-UNITEN, 43000 Kajang, Selangor, Malaysia; nowshad@uniten.edu.my

**Keywords:** diabetes, glucose, metamaterial, sensors

## Abstract

This paper presents an oval-shaped sensor design for the measurement of glucose concentration in aqueous solution. This unit cell sensing device is inspired by metamaterial properties and is analytically described for better parametric study. The mechanism of the sensor is a sensing layer with varying permittivity placed between two nozzle-shaped microstrip lines. Glucose aqueous solutions were characterized considering the water dielectric constant, from 55 to 87, and were identified with a transmission coefficient at 3.914 GHz optimal frequency with double negative (DNG) metamaterial properties. Consequently, the sensitivity of the sensor was estimated at 0.037 GHz/(30 mg/dL) glucose solution. The design and analysis of this sensor was performed using the finite integration technique (FIT)-based Computer Simulation Technology (CST) microwave studio simulation software. Additionally, parametric analysis of the sensing characteristics was conducted using experimental verification for the justification. The performance of the proposed sensor demonstrates the potential application scope for glucose level identification in aqueous solutions regarding qualitative analysis.

## 1. Introduction

In recent years, one of the most common diseases, diabetes, has taken the lead in terms of worldwide death rate. In 2005, more than 1.1 million people died as a direct result of diabetes. Statistically, approximately 8.8% of the global adult population was suffering from this disease in 2017 and by 2045, this will rise to 9.9%. The World Health Organization (WHO) indicates [1] that more than 220 million people currently live with diabetes. One of the critical issues that arises is problems with different physical organs or diseases such as coronary artery block, peripheral arterial and cerebrovascular disease associated with diabetes due to lack of discipline in the patient’s daily routine. Additionally, patients gradually lose their hair and develop skin issues. With time, the number of medications patients take increases, which incurs immune system degradation. At a certain stage, normal body kidney function deteriorates and the dialysis procedure starts. As a result, bacterial infection, organ failure, etc., propagate over time and patients expire within a few years [2]. The existing method of managing diabetes is dependent on taking medication, insulin, and measuring blood glucose levels using a portable device, onto which blood taken from pricking the patient’s finger is placed. Using invasive glucose sensors means that patients have to prick their finger for a drop of blood multiple times a day, about 1800 times per year, which also involves a higher risk of infection. For these reasons, in the last few years, non-invasive or nominally invasive methods of blood glucose measurement techniques have attracted the attention of different researchers. Some products now even include novel solutions such as placing a device on the patient’s belly or finger and within few seconds, glucose levels are indicated on the display. However, these products are still under development or are not widely available for commercial use.

Non-invasive or nominally invasive measuring techniques of glucose concentration in blood are of great interest among researchers around the world. Nevertheless, a large-scale study has proven that, with constant blood glucose monitoring, the patient can avoid any complications. The precise and regular knowledge of the blood glucose level is consequently mandatory and, in conjunction with appropriate treatments, levels must be maintained in the range of 70 to 120 mg/dL [3]. Currently, main glucose monitoring techniques, which are based on electrochemical reactions, have demonstrated high accuracy and strong reliability [4]. However, these solutions lead to the destruction of blood samples. 

Consequently, major efforts are being made to develop techniques to measure the glucose concentration in blood non-invasively. Different electromagnetic liquid sensors have been developed for fluid characterization [5,6,7,8], for bio-liquids analysis [9,10,11], and more specifically for glucose monitoring applications [12,13,14], and they have demonstrated that microwaves are appropriate for aqueous solution analysis. These sensors are either based on cumbersome resonant structures, more sensitive to glucose variation, or too miniature for on-chip system integration [13,14], thus requiring further resolution (both sensitivity and repetitiveness) improvements. The fact is that metamaterial has a unique characteristic though it is engineered using naturally available materials. The anisotropic magnetic metamaterial (which means large permeability in one direction, with very low permeability in the perpendicular direction) has the property to distinctively respond to magnetic resonance concerning sensitivity [15].

Moreover, conventional sensitivity in glucose sensing has size issues, whereas metamaterial-inspired sensors can be miniature. Another research group [16] reported air-bridge enhanced capacitor-based glucose detection using complex permittivity characterization. They have also extracted all parameters form the measured S-parameter. The study was conducted on different subjects and was found to have a detection resolution of 0.61 MHz/mg dL^−1^ to 3.4 pF/mg dL^−1^. Excellent linearity of correlation makes the procedure more reliable regarding analysis. A robust reusable glucose level sensor was reported in [17], where the proposed glucose biosensor chip exhibits linear detection ranges with high sensitivity at the center frequency. It has been claimed to have a sensitivity of up to 199 MHz/mg mL^−1^ with a repose time that is less than 2 s. This dynamic Radio Frequency (RF) resonator-based biosensor has a multi-dimensional detection capacity estimated from the measured S-parameter. 

This paper presents the experimental demonstration of a miniature microwave sensor, which may be envisioned for glucose monitoring in aqueous solution without compromising its sensitivity and resolution, and its application as a potential non-invasive blood glucose analysis method. 

## 2. Design Methodology 

The electromagnetic response of the material changed according to the incidence of a wave, and two fundamental parameters govern this change, namely, permittivity (*ε*) and permeability (*μ*). These are a material’s intrinsic properties; they are unique for every material owing to singular atomic configurations [18]. The properties of a metamaterial (MM) structure behave differently from those of its constituent materials, where each unit cell individually behaves as an electric dipole. MM structures simultaneously exhibit negative permittivity and permeability. The principle of operation of MM-inspired sensors is based on the change in reflection/transmission coefficients (scattering parameters, S11, S21), which is induced in the sensed parameters (dielectric change) owing to variations in the permittivity, permeability, or refractive index of the MM-based resonator. 

This study proposes a metamaterial biosensor consisting of an oval-shaped patching with sandwiched T strips inside. Also, a nozzle patch connects the round sensing area of 0.25 mm radius. The sensing area is simply a cavity in the dielectric substrate and the purpose is to hold the sample as well as forming a capacitance dependent on the amount of glucose. The unit cell consists of a Rogers RO4350B substrate of 1.575 mm thickness with a 0.035 mm copper patch layer. The geometrical dimension details are shown in Table 1. RO4350B materials are proprietary woven glass reinforced hydrocarbon/ceramics with an electrical performance close to PTFE/woven glass and the manufacturability of epoxy/glass providing tight control on the dielectric constant and low loss while utilizing the same processing method as standard epoxy/glass. 

At first, the microstrip transmission line model was analyzed to design the unit cell sensor because the inductance (L) and capacitance (C) formed for patching were specifically dependent on lumped components for modeling the prototype. Lumped elements are especially suitable for broadband hybrid microcircuits where the *Q* factor, a smaller size and lower costs are of prime importance. The proposed design was created with consideration to such issues according to the study in [19], along with a comprehensive mathematical model developed in the presence of ground planes, proximity effects, fringing fields, parasitics, etc. Optimization and the necessary formulation of the microstrip patches with regard to the lumped elements of the design carried on the relationship between the wavelength and operating frequency followed in [19,20]. Furthermore, these components show notable characteristics regarding the values and dependency with low *Q* factor and the resonance frequency of S-parameters. As shown in Figure 1a, the transmission line model and our design intended to achieve an oval or semi-circular shape with higher *Q* and higher inductance values. Similarly, for the microstrip circuit capacitance formed by the conductor patch on the dielectric substrate, low values are preferred due to practical considerations like per unit area capacitance. Hence, let us numerically determine the value of the total equivalent inductance and capacitance suggested in reference [19]
(1)$$L(\mathrm{nH})=1.257\times 10^{-3}a[\ln(\frac{a+b}{w+t})+0.078]K_{g}$$
(2)$$C(\mathrm{pF})=\frac{10^{-3}\varepsilon_{rd}wl}{36\pi d}$$
where,
*a* = major axis length*b* = minor axis length*l* = length of the micro strip line*w* = width of the micro strip line*t* = thickness of the microstrip line$\varepsilon_{rd}$ = dielectric constant of dielectric film*d* = average distance between mutual striplinescorrection factor = $K_{g}=0.57-0.145\ln\frac{w_{s}}{h^{\prime }}$*w_s_* = substrate width$h^{\prime }$ = substrate thickness

Using Equations (1) and (2), lumped inductance and capacitance is 3.439 × 10^−5^ nH and 1.933 × 10^−5^ pF. Then, the design and simulation are performed through the commercially available Computer Simulation Technology (CST) Microwave studio 2017 software. Figure 1 shows the proposed design with a major dimension of unit cell biosensor. 

## 3. Results and Discussion

### 3.1. E-Field, H-Field and Surface Current Analysis

EM field distribution (Figure 2) at the resonance frequency of the transmission coefficient has to be explained from a physical and mathematical point of view. As we know, according to ‘Helmholtz equation’, the E-field and the H-field are both dependent on the propagation constant, γ, of the medium. Whereas, this γ has frequency-dependent characteristics, including the permittivity, permeability, and conductivity of the material under consideration (Rogers RO4350B). Moreover, the tangent loss is 0.0037 and the dielectric constant, *ε_r_*, is 3.48. Now, the transmission line equivalent circuit (Figure 1a) clearly shows consecutive capacitance formation along two edges of the oval shape. Thus, at the resonance frequency (3.914 GHz), these capacitors create a strong electric field by following the linear homogeneous differential equation derived from the ‘Helmholtz equation’ since the above parameters have a non-linear relationship with propagation constant, γ. A similar solution applies for H-field distribution (Figure 2b) at resonance frequency because the magnetic field component’s mutual coupling distributes its energy according to the homogeneous solution of the field equation. Moreover, the vertical split gap forms additional LC resonance to accelerate H-field formation. 

The surface current distributions shown in Figure 3 have significant distribution with high density observed as resonance frequency. At 3.914 GHz, outer and inner resonators have a significant amount of surface current roaming around the edges. An important point to remember is that the microstrip pattern uses Cu (thickness of 0.035 mm) on top of the RO4350B substrate. Therefore, a traveling wave may face a situation where, at low frequency, the wave has a good conductor medium but with increasing frequency, the dielectric property of the same medium decreases. Additionally, the vertical and horizontal portion of the unit cell has more area compared to other portions. As a result, the conduction current density and the displace current density both contribute to a higher resultant density current. However, the loss tangent value becomes higher as charges travel through the oval-shaped wall. An interesting point to note is that at the nozzle point, where the sensing layer is placed, a significant number of charge elements traveling back and forth can be seen. Due to the narrower split gap, an overlapping charge distribution may hamper the uniform charge flow and show such densely surface current. Though the “hot” area shown in Figure 3 does not have any significant current element, the sandwiched T strip does have a certain amount of surface current element which is more than 24 dB. Hence, during sample injection in the sensing area, this can induce a change in capacitance (C4) and help to identify the parametric changes for glucose sensing in the proposed design. 

### 3.2. Transmission and Reflection Performance

Herein, we designed and analyzed the metamaterial biosensor according to the finite integration technique (FIT)-based simulation method in CST microwave studio regarding the basic transmission and reflection coefficient. After design completion, the boundary condition was applied on waveguide port 1 and 2 without imposing any polarization angle. For optimum results from simulation, material properties such as permittivity and permeability, as well as thermal and electrical properties were kept at normal conditions. The unit cell boundary condition was given an electric field in the X-axis and a magnetic field in the Y-axis, whereas the Z-axis was kept as an open space for the propagation of the field. For the time being, the split gap remained 1.5 mm and this gap change effect will be discussed in this section. The reflection (S11) and the transmission (S21) coefficient are scattering parameters to identify the characteristics of this design sensor concerning Electromagnetic (EM) field interactions within the targeted frequency range. 

Figure 4a,b depict that the transmission coefficient has −0.603 whereas the reflection coefficient is +0.296 at resonance frequency 3.914 GHz. Figure 4c,d show the Smith chart of transmission coefficient with a normalized impedance of 376.7 and 50 ohm. Starting from 2 GHz, the inductive property of line sweeping, with an increasing frequency and at 3.914 GHz, exhibited a quarter wave LC resonance. For broadband response, optimizing the match at a single frequency can be suboptimal across a band. After that, the transmission line gives capacitive reactance at 5 GHz with 376.7 ohms while remaining inductive for the 50-ohm normalized impedance line. This change will indicate whether we have a sensing ability or not through sensing layer permittivity change. In glucose measurement, permittivity depends on its dielectric constant. Thus, we intend to change layer *ε_r_* and observe resonance frequency shifting. Since we already know the s-parameters from simulation, let us numerically analyze the designed sensor using the refractive index parameter. To do so, we can choose the Nicolson-Ross-Weir (NRW) method, Transmission-Reflection (TR) method or Direct-Refractive Index (DRI) method. However, as we consider the simple unit cell, the best choice is the DRI method [21] because the previous two methods are more convenient for metamaterials that are composed of metallic arrays with complex shapes. Also, the TR method requires material impedance to determine and correct the branch index with a trigonometric function. So,
(3)$$\eta \approx \frac{c}{j\pi fd}{\left\{\frac{{\left(S_{21}-1\right)}^{2}-S_{11}{}^{2}}{{\left(S_{21}+1\right)}^{2}-S_{11}{}^{2}}\right\}}^{1/2}$$

Figure 5 indicate the basic properties (Figure 5a–c) of the sensor as well as absorbing capacity (Figure 5d) in simulated environment. All properties were extracted using S11 and S21 parameter. For clarification with regard to metamaterial-based sensors, it is noteworthy to mention that traditional microring resonators have lower sensitivity compared to DNG metamaterial since these types of materials amplify the evanescent wave [22]. Moreover, the resolution of sensitivity is also enhanced when using such metamaterials. Double negative (DNG) values of the unit cell at the resonance frequency are shown in Table 2. 

Hence, the DNG property and the surface current amount at the resonance frequency in the sensing layer region indicate the sensing potentiality of this proposed design because the variation of permittivity on the dielectric cavity shows resonance frequency shifting. However, the absorption loss of the proposed structure can be revealed using numerical computation as described in [23], for example, a photonic structure with background material, RO4350B, and infiltrated with blood glucose. The total absorption loss is due to blood and the substrate material. Thus, absorption can be written as
(4)$$Absorption_{blood}=e^{-\alpha d}$$
where *α* is called the absorption coefficient of blood at 3.914 GHz and *d* is the total diameter of blood containing the sample. Further,
(5)$$\alpha =\frac{4\pi }{\lambda }k$$
where *k* is an extension coefficient like $k=\sqrt{\varepsilon_{img}}$. 

Now that we have concluded that the metamaterial-inspired design sensor shows good material property in simulation, let us check its sensing capacity by varying sensing layer permittivity (55 to 87). We may assume that the water layer is the glucose sample and the split gap remains the same. Also, the excitation criteria and boundary conditions of the unit cell sensor used earlier are the same, we are just adding a parametric sweep for permittivity with a step width of 8. Figure 6 illustrates simulated performance which indicates that, starting from 55 to 87, there is a notable amount of shift on S21 which is, on average, a 0.037 GHz/step. An enlarged view of this has been added for better understanding and further study of such performance conducted on split gap variation to obtain a parametric change. The split gap changed from 1.00 to 1.50 mm and the S21 response is plotted in Figure 7. It is evident that mutual capacitance variation has a direct impact on performance, showing a gradual decrease of shifting value as the split gap increases. To some extent, it is non-linear, but overall, it is a linear change. 

Certainly, a Phase Locked Loop (PLL) can resolve such resonance frequency shifting behavior. However, using PLL in the case of wearable sensing element or fluid-detection or bio-sensing would be a size factor for the development of a device. Also, in most cases, bio-sensing requires wider bandwidth operation where PLL might not perform well. Hence, making the sensor more robust with precise sensing ability enhancements, metamaterial-inspired glucose sensing can be a potential alternative compared to conventional devices. In Table 3, we present a comparative study for a general idea of this metamaterial-inspired glucose sensor. Although some prototypes showed precise detection values, this design intends to achieve this sensing ability with an optimum smaller size using Rogers RO4350B substrate and ensure that the metamaterial property is in the focus spectrum. 

### 3.3. Sensitvity in the Experimental Setup

To verify the theoretical results, an experimental setup was designed as shown in Figure 8 with the fabricated unit cell in the presence of glucose in the sensing area and Vector Network Analyzer (VNA) Agilent N5227A. We took human serum from four different samples with glucose concentrations ranging from 68 to 150 mg/dL. With the help of an injection needle, the serum was poured one drop at a time on the sensing area where each drop contains approximately 30 mg/dL of the solution, and a maximum of four drops were tested on the area. 

In Figure 8b,c, measured and simulated transmission co-efficient responses are plotted. These plots show approximately the same responses between theoretical and experimental setups. However, there is a sudden resonance at 4.26 GHz because the solution pouring on the sensing area was difficult to position in the specific area. Nevertheless, controlling the injection amount per drop (four drops), different measured responses are plotted in Figure 8c; this indicates that the unit cell has sensing capacity. Approximately 0.037 GHz shifted/(30 mg/dL) glucose solution was added on the sensing area. 

## 4. Conclusions

This oval-shaped metamaterial sensor design presented for glucose level sensing in a nominally invasive way is validated with experimental analysis. Simulated and analytical data indicate potential sensing capacity in the S-band because the 0.037 GHz/step width of permittivity change is a notable variation. Additionally, it exhibits DNG property with slight absorption characteristics which make the proposed design more appealing for biosensing applications. Despite simulation, fabrication procedures have been conducted for more accurate results verification. Overall, metamaterial-based glucose measurement has great potential in the biosensing application field. 

## Figures and Tables

**Figure 1 sensors-19-00169-f001:**
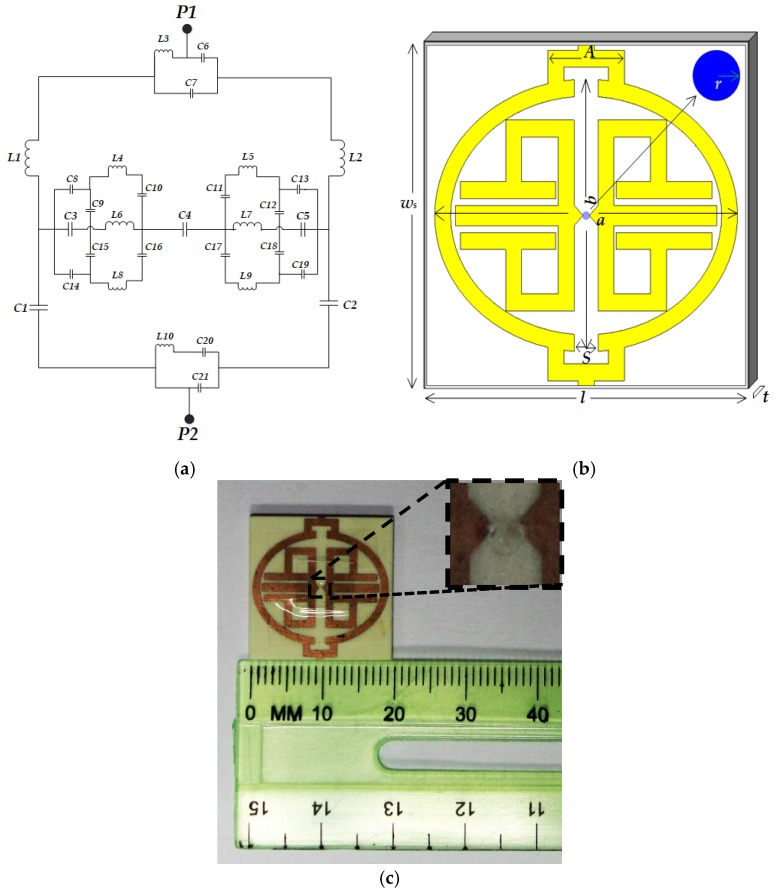
Metamaterial sensor design. (**a**) Equivalent circuit of unit cell design; (**b**) Unit cell with major dimensions; (**c**) Fabricated cell with sensing point in inset.

**Figure 2 sensors-19-00169-f002:**
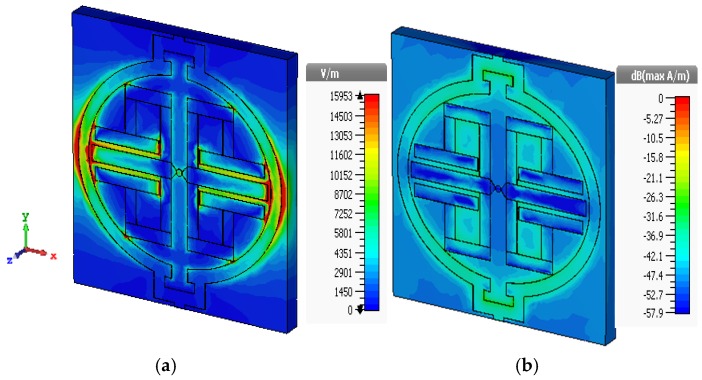
At resonance frequency (3.914 GHz), the (**a**) E-field distribution; (**b**) H-field distribution.

**Figure 3 sensors-19-00169-f003:**
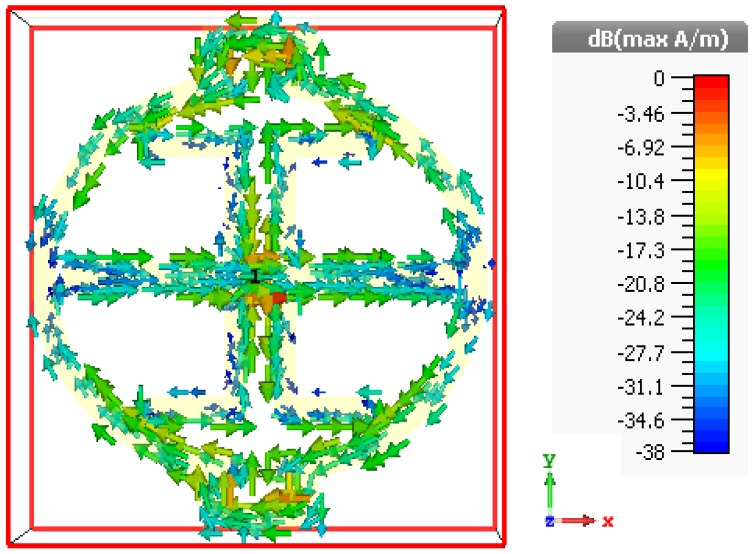
Surface current distribution at resonance frequency 3.914 GHz.

**Figure 4 sensors-19-00169-f004:**
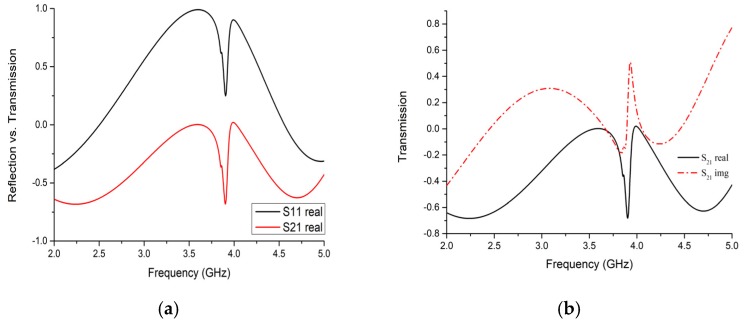
Reflection and transmission properties in the simulated design. (**a**) Reflection vs transmission; (**b**) Transmission characteristics; (**c**,**d**) Transmission characteristics using the Smith chart of the unit cell along propagation (normalized impedance 376.7 and 50 ohms).

**Figure 5 sensors-19-00169-f005:**
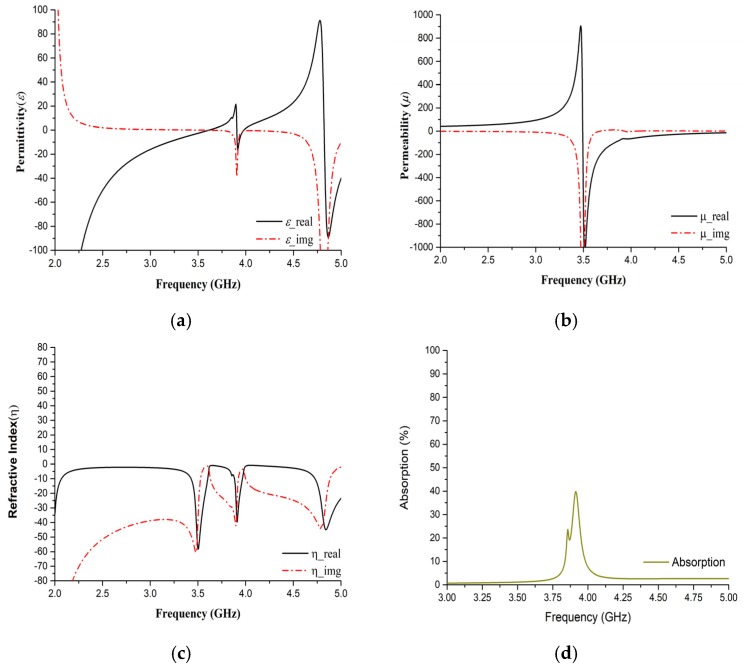
Simulation performance of the proposed sensor with regard to (**a**) Permittivity (*ε*); (**b**) Permeability (*µ*); (**c**) Refractive index (*η*); (**d**) Absorption with the focused frequency spectrum.

**Figure 6 sensors-19-00169-f006:**
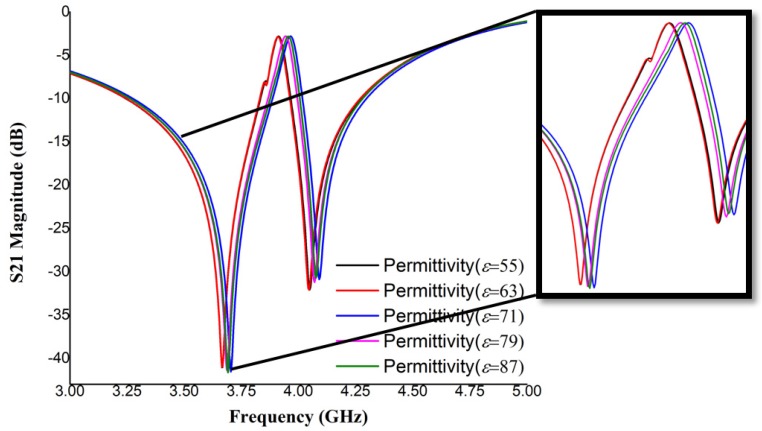
Transmission coefficient performance (simulation) regarding the sensing varying layer dielectric constant (with the shifting portion in the enlarged view).

**Figure 7 sensors-19-00169-f007:**
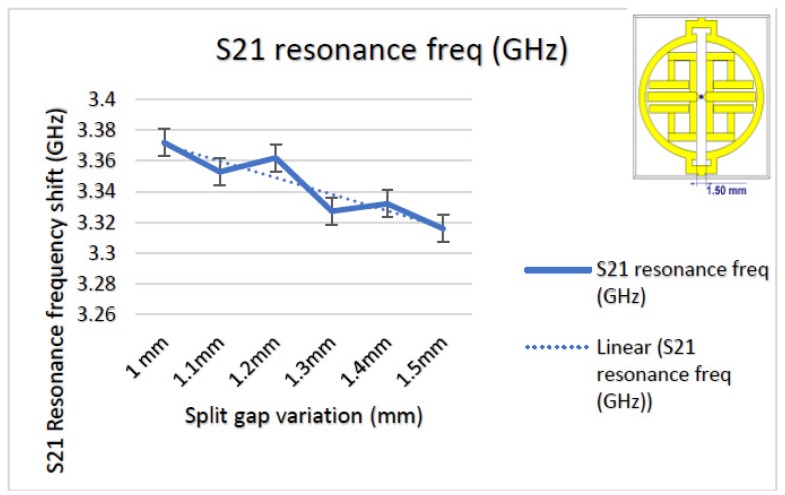
Transmission coefficient (S21) performance with split gap variation.

**Figure 8 sensors-19-00169-f008:**
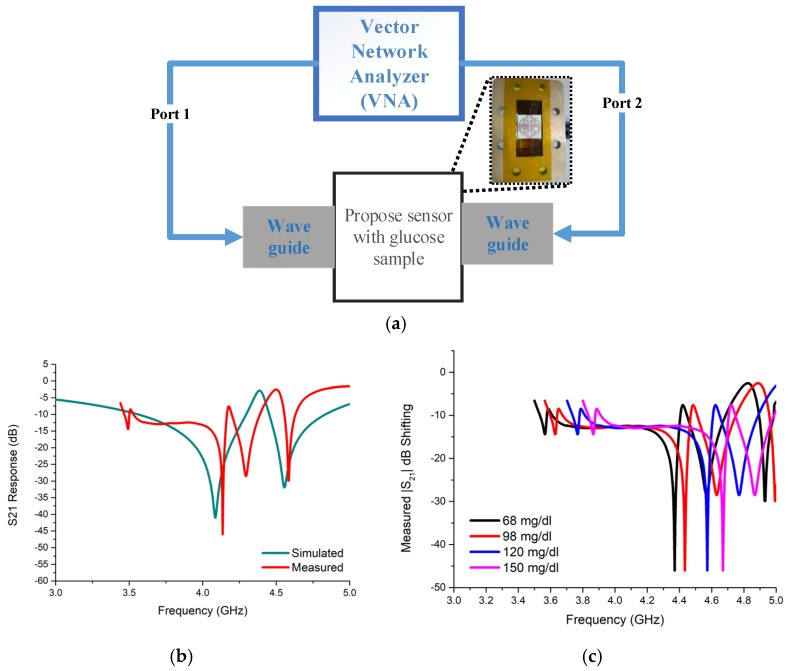
(**a**) Experimental setup for sensing ability measurement. (**b**) S21 response of simulated and measured unit cell glucose sensor; (**c**) Measured S21 resonance variation with different glucose concentrations.

**Table 1 sensors-19-00169-t001:** Oval-shaped metamaterial sensor.

Parameter	*l*	*w_s_*	*a*	*b*	*r*	*S*	*A*
**Size (mm)**	20	20	19	16	0.25	1.5	4.80

**Table 2 sensors-19-00169-t002:** The double negative (DNG) property of the metamaterial-inspired glucose sensor.

Resonance Frequency (GHz)	Permittivity, *ε*	Permeability, *µ*	Reflection Co-efficient, *η*
3.914	−16.55	−65.40	−38.12

**Table 3 sensors-19-00169-t003:** A comparative study of glucose sensors with the proposed design.

Ref. #	Size (mm)	Substrate Material	Operating Frequency (GHz)	Application Procedure	Sensitivity	Remarks
[24]	20 × 5.55	GML 1000	1–2	Non-invasive	10 mmol/L	Artificial transmission line used for monitoring
[2]	75 × 50	Inductive coil based	1–2	Non-invasive	400 mg/dL	Data fusion prediction and accuracy enhancement
[25]	Sensor unit (20 × 15)	SU-8	0–55	Non-destructive and Quantitative	7.6 × 10^−3^ dB/(g/L)	Thin film microstrip technique
[26]	80 × 2.4485	Rogers RT 5880	0–2.4	Non/minimally invasive	−40 dB/2.5 mL	High *Q* sensor
[27]	40 × 20	FR-4	1–3	Invasive	Range 20–100 mg/mL	Microwave filter as Sensor device
[28]	20 × 40	Al_2_O_3_	1–6	Non-invasive	21 dB shift/(50 mg/dL)	Modified Hilbert curve
[29]	20 × 15	Rogers RT 5880	1–2	Glucose Sensing	1.6 MHz shift/(1–15 g/dL)	SRR resonator without metamaterial
Proposed Sensor	20 × 20	Rogers RO4350B	2–5	Non-invasive	0.037 GHz shift/(30 mg/dL)	Sensing capacity with DNG property & Minimally absorption

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
