# Peer review of "Left-Handed Metamaterial-Inspired Unit Cell for S-Band Glucose Sensing Application"

_sensors, 2019, doi:10.3390/s19010169_

Round 1
Reviewer 1 Report
Dear authors,
thank you for your nice work.
A general remark, it does not get clear why the Metamaterial approach raises sensitivity or in other words induces a larger shift of resonance frequency than e.g. detuning a ring resonator like presented in literature. It would have been good to compare sensitivity with a unified metric. Your table 2 does not fully disclose your advancements. You relate to permitivitty changes whereas others relate to glucose concentration changes. That makes it difficult to compare your work to state of the art.
Furthermore I am missing a discussion on sensing area. If I look at fig. 3, I see three "hot" areas.
The green areas are 20 dB down, which means they not really contribute to sensing area.
From other work in glucose sensing I would have expected a fair comparison to planar ring resonators or split ring resonators.
Why is the DNG behaviour key for raising sensitivity? In Fig. 6 a sharp resonance around 3.9 GHz Shows up. Also its frequency shift is shown. You do not explain why your structure delivers this peak.
Regarding frequency shift behaviour it is key to know how much the peak shifts within the range of physiological relevant glucose ranges. A PLL can easily resolve 25 kHz. So even if frequency shift is low, this does not question your achievements.
My suggestion would be to leave out the section 3.3 on regression and focus on sensitivity and sensing area. To obtain reliable measurements it is key to average over a sensor area to cope with locally limited glucose increase. With the earlier work by other groups sensing would happen e.g. only at the slot of sloted ring resonator. On the opposite, your structure is wider spread and may offer possibilities for a hot area rather than a single hot spot as shown in literature so far.
English phrasing, especially sentence order and passive wording needs revisit.
Thanks
Reviewer
Author Response
Dear Reviewer,
The authors would like to thank the anonymous reviewer for his effort in reviewing the manuscript and for his valuable and constructive comments and fruitful observations that helped in improving the quality of the manuscript to a publishable standard.
Thank you again.

Reviewer 2 Report
You can find reviews comments in the attached file.

Author Response

(The authors gave the same response as above.)

Round 2
Reviewer 1 Report
Thanks for incorporating comments. I wish you nice xmas season. / Reviewer
Author Response
Dear Reviewer,
The authors would like to thank the anonymous reviewer for his effort in reviewing the manuscript and for his valuable and constructive comments and fruitful observations that helped in improving the quality of the manuscript to a publishable standard.
Thanking you again for your technical suggestion and wishing you a very happy ‘Christmas Day’ and ‘Happy New Year 2019’.
Reviewer 2 Report
1. Put reference for Line 51. Your results are in mg/dl. So, modify this value to the same unit.
2. Review question 4.1 and reply look different.
Author Response
Dear Reviewer,
The authors would like to thank the anonymous reviewer for his effort in reviewing the manuscript and for his valuable and constructive comments and fruitful observations that helped in improving the quality of the manuscript to a publishable standard.
Detailed below are the responses (attached herewith) to the reviewer comments and suggestions. The corrections in the revised manuscript, marked in Green for Reviewer#2
